# RETHINKING NEURAL NETWORK QUANTIZATION

## ABSTRACT

Quantization reduces computation costs of neural networks but suffers from performance degeneration. Is this accuracy drop due to the reduced capacity, or inefficient training during the quantization procedure? After looking into the gradient propagation process of neural networks by viewing the weights and intermediate activations as random variables, we discover two critical rules for efficient training. Recent quantization approaches violates the two rules and results in degenerated convergence. To deal with this problem, we propose a simple yet effective technique, named scale-adjusted training (SAT), to comply with the discovered rules and facilitates efficient training. We also analyze the quantization error introduced in calculating the gradient in the popular parameterized clipping activation (PACT) technique. Through SAT together with gradient-calibrated PACT, quantized models obtain comparable or even better performance than their full-precision counterparts, achieving state-of-the-art accuracy with consistent improvement over previous quantization methods on a wide spectrum of models including MobileNet-V1/V2 and PreResNet-50.

## 1 INTRODUCTION

Deep neural networks have gained rapid progress in tasks including computer vision, natural language processing and speech recognition [Voulodimos et al. (2018); Young et al. (2018); Nassif et al. (2019)], which have been applied to real world systems such as robotics and self-driving cars [Lillicrap et al. (2015); Janai et al. (2017)]. Recently, it becomes a significant challenge to deploy the heavy deep models to resource-constrained platforms such as mobile phones and wearable devices. To make deep neural networks more efficient on model size, latency and energy, people have developed several approaches such as weight prunning [Han et al. (2015)], model slimming [Liu et al. (2017); Yu et al. (2018)], and quantization [Courbariaux et al. (2015; 2016)]. As one of the promising methods, quantization provides the opportunity to embed bulky and computation-intensive models onto platforms with limited resources. By sacrificing the precision of weights [Courbariaux et al. (2015); Han et al. (2015); Li et al. (2016); Zhu et al. (2016); Leng et al. (2018)] and even features [Courbariaux et al. (2016); Rastegari et al. (2016); Zhou et al. (2016; 2017); Park et al. (2017); Mellempudi et al. (2017); Mishra & Marr (2017); Mishra et al. (2017); Xu et al. (2018); Choi et al. (2018); Jacob et al. (2018); Zhang et al. (2018); Zhuang et al. (2019); Gong et al. (2019)], model sizes can be shrunk to a large extent, and full-precision multiplication is replaced by low-precision fixed-point multiplication, addition or even bitwise operations, requiring much reduced latency and energy consumption during inference.

The quantized models are either directly derived from full-precision counterparts [Sheng et al. (2018); Krishnamoorthi (2018)], or through regularized training [Rastegari et al. (2016); Choi et al. (2018); Bai et al. (2018)]. Training-based methods better exploit the domain information and normally achieve better performance. However, the quantized models still suffer from significant accuracy reduction. To alleviate this problem, instead of manually designing the number of bits in different layers, recent works resort to automated search methods [Elthakeb et al. (2018); Wu et al. (2018); Wang et al. (2019); Uhlich et al. (2019); Lou et al. (2019)]. Despite all the effort to improve quantized models, we notice that a fundamental problem of network quantization remains untouched, that is, whether accuracy reduction is due to the reduced capacity of quantized models, or is due to the improper quantization procedure. If the performance is indeed affected by the quantization procedure, is there a way to correct it and boost the performance?

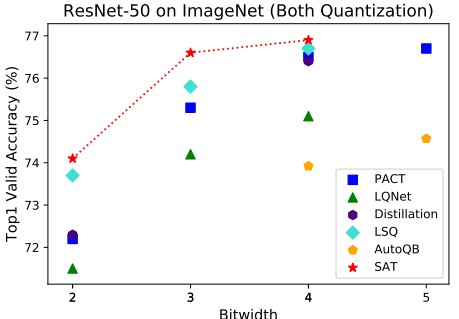 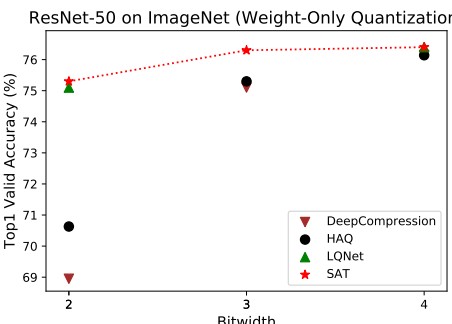

Figure 1: Comparison of approaches with ResNet50 on ImageNet dataset under different quantization levels. Left: Quantization on both weight and activation. Right: Weight-only quantization. The bit-width represents equivalent computation cost for mixed-precision methods (AutoQB and HAQ).

In this paper, we first investigate the condition of efficient training of a generic deep neural network by analyzing the variance of the gradient with respect to the *effective weights* involved in convolution or fully-connected operations. Here, effective weights denote the weights directly involved in convolution and other linear operations, which can be transformed weights in either quantized or unquantized training. With some semi-quantitative analysis (together with empirical verfication) of the scale of logit values and gradient flow in the neural network during training, we discover that proper convergence of the network should follow two specific rules of efficient training. It is then demonstrated that deviation from these rules leads to improper training and accuracy reduction, regardless of the quantization level of weights or activations.

To deal with this problem, a simple yet effective technique named scale-adjusted training (SAT) is proposed, with which the rules of efficient training are maintained so that performance can be boosted. It is a general approach which can be integrated with most existing quantization-aware training algorithms. We showcase the effectiveness of the SAT approach on a recent quantization approach PACT [Choi et al. (2018)]. On the other hand, we also discover the quantization error introduced in calculating the gradient with respect to the trainable clipping level in the PACT technique, which also degenerates accuracy, especially for low precision models. A schedule with calibrated gradient for PACT, namely CG-PACT, is proposed to correct the inaccurate gradient. With the proposed SAT and CG-PACT, we achieve state-of-the-art performance of quantized neural networks on a wide range of models including MobileNet-V1/V2 and PreResNet-50 on ImageNet dataset, with consistent improvement over previous methods [Han et al. (2015); Choi et al. (2018); Zhang et al. (2018); Zhuang et al. (2019); Wang et al. (2019); Lou et al. (2019); Esser et al. (2019)] under the same efficiency constraints. As an example, Fig. 1 compares our method with some recent quantization techniques applied to ResNet-50 on ImageNet classification task. We believe our analysis and approaches will shed light on some existing issues of network quantization, and facilitate more advanced algorithms and applications.

## 2 RELATED WORK

**Uniform-precision quantization.** Quantization of deep models has long been discussed since the early work of weight binarization [Courbariaux et al. (2015; 2016)] and model compression [Han et al. (2015)].The majority of previous methods enforce the same precision for weights/activations in different layers during quantization. Early approaches focus on minimizing the difference in values [Rastegari et al. (2016)] or distributions [Han et al. (2015); Zhang et al. (2018)] between quantized weights/activations and full-precision ones. Recently, [Zhang et al. (2018)] proposes a learning-based quantization method, where the quantizer is trained from data. Regularizer for quantization is also proposed to implement binarized weights [Bai et al. (2018)]. Ensemble of multiple models with low precision has also been studied [Zhu et al. (2019)], demonstrating improved performance than single model under the same computation budget. [Esser et al. (2019)] proposes a quantizer with trainable step size, and improves training convergence by balancing the magnitude of step size updates with weight updates, based on some heuristic analysis. However, this method

focuses on training the step size, and scales the gradients, instead of analyzing the impact of model weights themselves on the training dynamics. Previous work have paid little attention to the impact of quantization procedure on training dynamics, and it remains unclear if previous algorithms are efficient in training during quantization procedure.

**Mixed-precision quantization.** Recent work attempts to use mixed-precision in one model, and weights and activations in different layers are assigned different bit-widths, resulting in better trade-offs between efficiency and accuracy of neural networks. Towards this end, automated algorithms are adopted to determine the most appropriate bit-width for each layer. Reinforcement learning are adopted to search bit-width configurations with guidance from memory and computation cost [Elthakeb et al. (2018); Lou et al. (2019)] or latency and energy produced by hardware simulators [Wang et al. (2019)]. [Wu et al. (2018)] and [Uhlich et al. (2019)] apply differentiable neural architecture search method to efficiently explore the searching space. Although these methods result in more flexible quantization architectures, the training procedure of quantized models still follows paradigms of uniform-precision quantization.

**Efficient Training.** Training dynamics of neural networks has been extensively studied, both on the impact of initialization schemes [He et al. (2015)] and on the convergence condition with mean field theory [Poole et al. (2016)]. Early work mainly focuses on networks with simple structure, such as with only fully-connected layers [Schoenholz et al. (2016)]. More advanced topics including residual networks and batch normalization are discussed with more powerful tools [Yang & Schoenholz (2017); Yang et al. (2019)], facilitating efficient training of CNN with 10k layers through orthogonal initialization of kernels [Xiao et al. (2018)]. From a different angle of view, [Zhang et al. (2019)] proposes a properly rescaled initialization scheme to replace normalization layers, which also enables training of residual network with a huge number of layers. These approaches mainly focus on weight initialization methods to facilitate efficient training. We extend this analysis to the whole training process, and discovered some critical rules for efficient training. These rules guide our design of an improved algorithm for network quantization.

## 3 EFFICIENT TRAINING OF NETWORK QUANTIZATION

### 3.1 BASIC QUANTIZATION STRATEGY

Historically, quantization of neural networks follows different conventions and settings [Krishnamoorthi (2018)]. Here we describe the convention adopted in this paper to avoid unnecessary ambiguity. We first train the full-precision model, which is used as a baseline for the final comparison. For quantized models, we use the pretrained full-precision model as initialization, and apply the same training hyperparameters and settings as full-precision model (including initial learning rate, learning rate scheduler, weight decay, the number of epochs, optimizer, batch size, etc.) to fine-tune the quantized model. For the input image to the model, we use unsigned 8bit integer (uint8), which means we do not apply standardization (neither demeaning nor normalization). Previous work sometimes avoid quantizing the first and last layers due to accuracy drop [Zhou et al. (2016); Wu et al. (2018)]. We follow a more practical setting to quantize weights in both layers with a minimum precision of 8bit [Choi et al. (2018)] in our main results. To investigate the effect of quantization level in these two layers, some additional results are shown in Appendix E. The input to the last layer is quantized with the same precision as other layers. As a widely adopted convention [Choi et al. (2018); Wang et al. (2019)], bias in the last fully-connected layer(s) and the batch normalization (BN) layers (including weight, bias and the running statistics) are not quantized, but as shown in Appendix G, batch normalization layers of most networks can be eliminated except for a few cases. Note that no bias term is used in convolution layers.

### 3.2 RULES OF EFFICIENT TRAINING

Before detailed analysis, we first formulate the problem we will investigate. Following [He et al. (2015); Schoenholz et al. (2016)], we will discuss a simplified network composed of repeated blocks, each consisting of a linear layer (either convolution or fully-connected) without bias, a BN layer, an activation function and a pooling layer. The last layer is a single fully-connected layer without bias, which produces the logit values for the loss function.

Suppose for the $l$-th block, the input is $x^{(l)}$, and the effective weight of the fully-connected layer is $\Xi^{(l)}$, which is called the effective weight because it is the de facto weight involved in forward and backward propagation of the network. In other words, we have

$$z_i^{(l)} = \sum_j \Xi_{ij}^{(l)} x_j^{(l)} \tag{1a}$$

$$y_i^{(l)} = \gamma_i^{(l)} \frac{z_i^{(l)} - \mu_i^{(l)}}{\sigma_i^{(l)}} + \beta_i^{(l)} \tag{1b}$$

$$\widetilde{x}_i^{(l+1)} = h(y_i^{(l)}) \tag{1c}$$

$$x_i^{(l+1)} = \frac{1}{(k_l^{\mathrm{pool}})^2} \sum_{p \in \mathcal{P}} \widetilde{x}_p^{(l+1)} \tag{1d}$$

where $l = 1, \ldots, L$, $L$ is the depth of the network, $\sigma$ and $\mu$ stand for running statistics, and $\gamma$ and $\beta$ denote the trainable weights and bias in the BN layer, respectively. $h(\cdot)$ is the activation function, which is assumed to be quasi-linear, such as the ReLU. $k_l^{\mathrm{pool}}$ is the kernel size of the pooling layer, which can be set to $1$ for the usual case with no pooling, and $\mathcal{P}$ is the set of pooling indices. Note that for the last layer $l = L$, only Eq. (1a) applies, and $z^{(L)}$ are the logits for calculating the cross entropy loss.

Optimization of the cross entropy loss is highly related to the scale of weights in the last fully-connected layer. Suppose the input to this layer is normalized with BN layer before pooling. To avoid saturation issue for calculating gradient, we have the following rule of efficient optimization for the cross entropy loss.

**Efficient Training Rule I (ETR I)** To prevent the logits from entering the saturation region of the cross entropy loss, the scale (measured by the variance) of the effective weight $\Xi^{(L)}$ in the last fully-connected layer should be sufficiently small. Specifically, we have

$$\mathbb{VAR}[\Xi_{ij}^{(L)}] \ll \frac{(k_{L-1}^{\mathrm{pool}})^2}{n_L} \tag{2}$$

where $n_L$ is the length of input features of the fully-connected layer, and $k_{L-1}^{\mathrm{pool}}$ is the kernel size of the preceding pooling layer ($k_{L-1}^{\mathrm{pool}} = 1$ if the preceding layer is not pooling). Empirically, we find a ratio less than $0.1$ is adequate.

Detailed derivation of this rule is given in Appendix A. Besides optimization of the cross entropy operation, training dynamics is also critical for efficient training of deep neural network. To analyze the training dynamics, we derive the following phenomenological law for gradient flowing during training.

**The Gradient Flowing Law** For the aforementioned network, the variances of gradients of loss with respect to the effective weights in two adjacent layers are related by

$$\mathbb{VAR}[\partial_{\Xi_{ij}^{(l)}} \mathcal{L}] = \kappa_1^{(l)} \cdot \frac{1}{(k_l^{\mathrm{pool}})^2} \cdot \frac{\widehat{n}_{l+1} \mathbb{VAR}[\Xi_{ij}^{(l+1)}]}{n_l \mathbb{VAR}[\Xi_{ij}^{(l)}]} \mathbb{VAR}[\partial_{\Xi_{ij}^{(l+1)}} \mathcal{L}] \tag{3}$$

On the other hand, if there is no batch normalization layer, we have

$$\mathbb{VAR}[\partial_{\Xi_{ij}^{(l)}} \mathcal{L}] = \kappa_2^{(l)} \cdot \frac{1}{(k_l^{\mathrm{pool}})^4} \cdot \widehat{n}_{l+1} \mathbb{VAR}[\Xi_{ij}^{(l+1)}] \mathbb{VAR}[\partial_{\Xi_{ij}^{(l+1)}} \mathcal{L}] \tag{4}$$

Here, both $\kappa_1^{(l)}$ and $\kappa_2^{(l)}$ are empirical parameters of order $\mathcal{O}(1)$ with respect to $n_l$. $\mathcal{L}$ is the loss function, and $n_l = c_l \cdot (k_l^{\mathrm{conv}})^2$ and $\widehat{n}_l = c_{l+1} \cdot (k_l^{\mathrm{conv}})^2$ represent the number of input and output neurons of the $l$-th layer, respectively. $c_l$ and $c_{l+1}$ denote the number of input and output channels, and $k_l^{\mathrm{conv}}$ is the kernel size of this layer (which can be set to $1$ for fully-connected layer).

Note that similar results have been studied for network initialization in [He et al. (2015)] and extensively studied with mean field theory [Schoenholz et al. (2016); Yang & Schoenholz (2017); Xiao et al. (2018)]. Here, we generalize the randomness assumption and empirically verify that the conclusion holds along the whole training procedure (see Appendix A). Note that viewing weights and

inputs as random variables along the whole training procedure is also adopted in previous literature of statistical mechanics approaches to analyze training dynamics of neural networks [Seung et al. (1992); Watkin et al. (1993); Martin & Mahoney (2017)].

To avoid gradient exploding/vanishing problems, the magnitudes of gradients should be on the same order when propagating from one layer to another. Suppose $k_l^{\mathrm{pool}} = 1$, that is, there is no pooling operation (since the number of pooling layers is much smaller than the number of linear layers, their effect can be ignored for simplicity). If we have BN after linear layer, then as long as the variance of weights in different layers are kept on the same order, the scaling factor in Eq. (3) will be $\mathcal{O}(1)$. Note that we have implicitly assumed that $n_l$ and $\widehat{n}_{l+1}$ do not differ in order. On the other hand, if there is no BN layer directly following the linear layer, the scale of weights should be on the order of the reciprocal of the number of neurons to keep the scaling factor in Eq. (4) on the order of $\mathcal{O}(1)$. Thus, we arrive at the basic rule for convergent training.

**Efficient Training Rule II (ETR II)** To keep the gradient of weights in the same scale across the whole network, either BN layers should be used after linear layers such as convolution and fully-connected layers, or the variance of the effective weights should be on the order of the reciprocal of the number of neurons of the linear layer ($n_l$ or $\widehat{n}_l$).

Detailed derivation of the gradient flowing law and ETR II is given in Appendix A.

With these rules, we are now ready to examine if training of quantized models is efficient. Following previous work PACT [Choi et al. (2018)], we use the DoReFa scheme [Zhou et al. (2016)] for weight quantization, and the PACT technique for activation quantization. These methods are popular and typical for model quantization, and it is noteworthy that they are adopted only to showcase the effectiveness of our approach, instead of limiting our analysis. Violations of training rules also exists in other quantization approaches such as BinaryNet [Courbariaux et al. (2015)], XNORNet [Rastegari et al. (2016)], Ternary Quantization [Zhu et al. (2016)], HWGQ [Cai et al. (2017)] as well as a differentiable quantization approach Darts Quant [Uhlich et al. (2019)], as weights distributions are changed during quantization/binarization procedure which results in potential violation of Efficient Training Rules. Thus we believe our approach is general and can be used to boost the performance of such quantization algorithms as well.

### 3.3 WEIGHT QUANTIZATION WITH SAT

The DoReFa scheme [Zhou et al. (2016)] involves two steps, clamping and quantization. Clamping transforms the weights to values between 0 and 1, while quantization rounds the weights to integers. We here analyze the impact of both steps on training dynamics.

#### 3.3.1 IMPACT OF CLAMPING

Before quantization, the weights are first clamped to the interval between 0 and 1. For a weight matrix $W$, we first clamp it to

$$\widetilde{W}_{ij} = \frac{1}{2} \left( \frac{\tanh(W_{ij})}{\max_{r,s} |\tanh(W_{rs})|} + 1 \right) \tag{5}$$

which is between 0 and 1. This transformation generally contracts the scale of large weights, and enlarges the difference of small scale elements. Thus, this clamping operation makes variables distributed more uniform in the interval $[0, 1]$, which is beneficial for reducing quantization error. However, the variance of the clamped weights will be different from the original ones, potentially violating the efficient training rules. Specifically, clamping leads to violation of ETR I in the last linear layer, and voilation of ETR II in linear layers without BN layers followed. For example, clamping on the full pre-activation ResNet [He et al. (2016)] and VGGNet [Simonyan & Zisserman (2014)] will both violate ETR II since they have linear layers without BN followed. In cases that convolution layers are followed by BN layers, the clamped weights are commensurate with each other and ETR II is preserved.

To understand the effect of clamping on ETR I, we first analyze a model using clamped weights without quantization, following the DoReFa scheme

$$\widehat{W}_{ij} = 2\widetilde{W}_{ij} - 1 \tag{6}$$

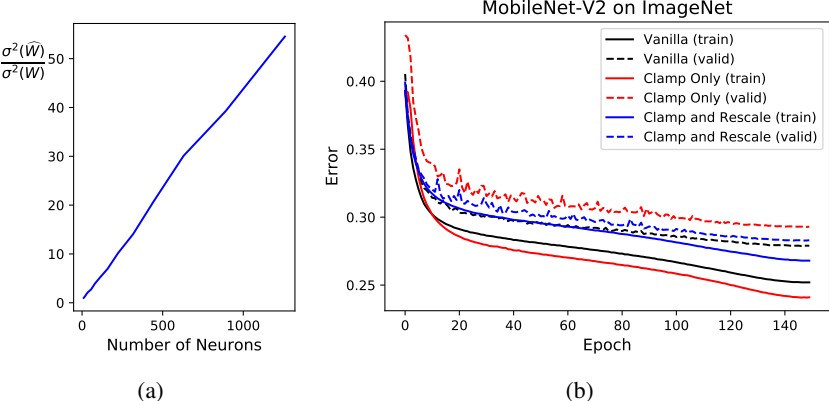

Figure 2: Effect of weight clamping. (a) The ratio of variances with respect to the number of neurons. Note that the plot is only a sampling result and different samples can give different results, but the order of magnitude remains meaningful. (b) Learning curves with different settings.

We have the effective weight $\Xi = \widehat{W}$ in this case. Fig. 2a gives the ratio between variances of the clamped and the original weights with respect to the number of neurons. As a common practice [He et al. (2015)], the original weights $W$ are sampled from a Gaussian distribution of zero mean and variance proportional to the reciprocal of the number of neurons. We find that for large neuron numbers, the variance of weights can be enlarged to tens of their original values, potentially increasing the opportunity of violating ETR I. Since the number of output neurons of the last linear layer is determined by the number of classes of data, we expect large dataset such as ImageNet [Deng et al. (2009)] to be more vulnerable to this problem than small dataset such as CIFAR10 (see Appendix A for more details).

To verify our analysis, we train the MobileNet-V2 on ImageNet using traditional method as well as using clamped weights, and compare their learning curves. As shown in Fig. 2b, clamping impairs the training procedure significantly, reducing the final accuracy by as much as 1%. Also, we notice that clamping makes the model more prone to the over-fitting issue, which is consistent with previous literature which claims that increasing weight variance in neural networks might worsen their generalization property [Seung et al. (1992); Watkin et al. (1993); Martin & Mahoney (2017)].

To deal with this problem, we propose a method named scale-adjusted training (SAT) to restore the variance of effective weights. We directly multiplies the normalized weight with the square root of the reciprocal of the number of neurons in the linear layer as in Eq. (7). Here $\mathbb{VAR}[\widehat{W}_{rs}]$ is the sample variance of elements in the weight matrix, calculated by averaging the square of elements in the weight matrix. In back-propagation, $\mathbb{VAR}[\widehat{W}_{rs}]$ is viewed as constant and receives no gradient. This simple strategy is named *constant rescaling* and works well empirically across all of the experiments. Note that here we have ignored the difference between weight variances across channels and just use variance of the weights in the whole layer for simplicity.

$$W_{ij}^* = \frac{1}{\sqrt{\widehat{n}\mathbb{VAR}[\widehat{W}_{rs}]}}\widehat{W}_{ij} \tag{7}$$

Fig. 2b compares the learning curves of vanilla method, and weight clamping with and without constant rescaling. It shows that SAT recovers efficient training and improves validation accuracy significantly after weight clamping. We also experiment with an alternative rescaling approach in Appendix C and notice similar performance. In the following experiments we will always use constant rescaling. For MobileNet-V2, we only need to apply SAT to the last fully-connected layer. For other models where convolution is not directly followed by BN such as full pre-activation ResNet [He et al. (2016)], we need to apply SAT to all such convolution layers (see Appendix E for more details). Note that the proposed method does not mean to be exclusive, and there are other methods to preserve the rules of efficient training. For example, different learning rates can be used for the layers with no BN layer followed. Before further discussion, we want to emphasize that

the clamping is only a preprocessing step for quantization and there is no quantization operation involved up to now.

### 3.3.2 IMPACT OF WEIGHT QUANTIZATION

With weights clamped to $[0, 1]$, the DoReFa scheme [Zhou et al. (2016)] further quantizes weights with the following function

$$q_k(x) = \frac{1}{a}\left\lfloor ax \right\rceil \tag{8}$$

Here, $\lfloor \cdot \rceil$ indicates rounding to the nearest integer, and $a$ equals $2^b - 1$ where $b$ is the number of quantization bits. Quantized weights are given by

$$Q_{ij} = 2q_k(\widetilde{W}_{ij}) - 1 \tag{9}$$

In this case, we have the effective weight given as $\Xi = Q$. To see the impact of quantization on the training dynamics, we compare the variance of the quantized weight $Q_{ij}$ with the variance of the full-precision clamped weight $\widehat{W}_{ij}$ in Appendix B. We find that for precision higher than 3 bits, quantized weights have nearly the same variance as the full-precision weights, indicating quantization itself introduces little impact on training. However, the discrepancy increases significantly for low precision such as 1 or 2 bits, thus we should use the variance of quantized weights $Q_{ij}$ for standardization, rather than that of the clamped weight $\widehat{W}_{ij}$. For simplicity, we apply constant scaling to the quantized weights of linear layers without BN by

$$Q_{ij}^* = \frac{1}{\sqrt{n_{\text{out}}\mathbb{VAR}[Q_{rs}]}}Q_{ij} \tag{10}$$

For typical models such as MobileNets and ResNets, only the last fully-connected layer needs to be rescaled, and such rescaling is only necessary during training for better convergence. For inference, the scaling factor (which is positive) can be discarded, with the bias term being modified accordingly, introducing no additional operations. For models with several fully-connected layers such as VGGNet [Simonyan & Zisserman (2014)], or with convolution layers not followed by BN layers, such as fully pre-activation ResNet, the scaling factors for these layers can be applied after computation-intensive convolutions or matrix multiplications, adding marginal computation cost.

### 3.4 ACTIVATION QUANTIZATION WITH CG-PACT

For activation quantization, [Choi et al. (2018)] proposes a method called parameterized clipping activation (PACT), where a trainable parameter $\alpha$ is introduced. An activation value $x$ is first clipped to the interval $[0, \alpha]$ with the hard $\tanh$ function, then scaled, quantized and rescaled to produce the quantized value $q$ as

$$\widetilde{x} = \frac{1}{2}\Big[|x| - |x - \alpha| + \alpha\Big] \tag{11a}$$

$$q = \alpha q_k\left(\frac{\widetilde{x}}{\alpha}\right) \tag{11b}$$

Quantized value obtained this way is called dynamic fixed-point number in [Mellempudi et al. (2017)] because $\alpha$ is a floating number.

Since $\alpha$ is trainable, we need to calculate gradient with respect to it besides the activation $x$ itself. For this purpose, the straight through estimation (STE) method [Bengio et al. (2013)] is usually adopted for backward propagation, which gives

$$q_k'(x) := 1 \tag{12}$$

As $q_k(x)$, the domain of definition for $q_k'(x)$ is also restricted to $[0, 1]$. Chain rule gives (see Appendix D for more details)

$$\frac{\partial q}{\partial \alpha} = \begin{cases} q_k\left(\frac{\widetilde{x}}{\alpha}\right) - \frac{\widetilde{x}}{\alpha} & x < \alpha \\ 1 & x > \alpha \end{cases} \tag{13}$$

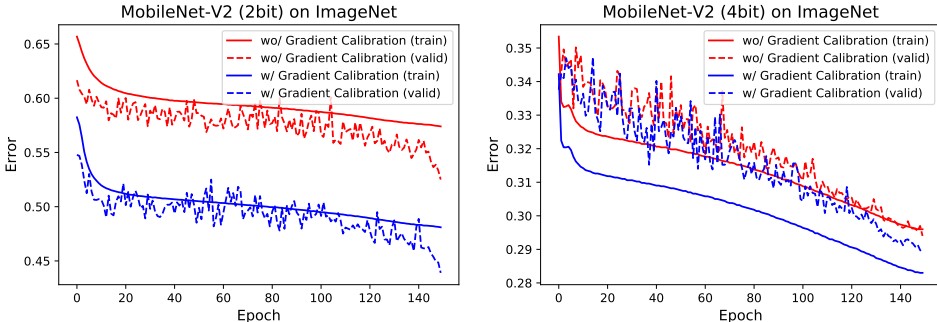

Figure 3: Effect of quantization error in calculting gradient for PACT. Left: 2bit MobileNet-V2 on ImageNet. Right: 4bit MobileNet-V2 on ImageNet.

Table 1: Comparison of quantization techniques with both weights and activation quantized.

| Quant. Method | Bit-widths | MobileNet-V1 | | MobileNet-V2 | |
|---|---|---|---|---|---|
| | | Acc.-1 | Acc.-5 | Acc.-1 | Acc.-5 |
| PACT | 4bits | 70.3 | 89.2 | 70.4 | 89.4 |
| HAQ | flexible | 67.40 | 87.90 | 66.99 | 87.33 |
| SAT (Ours) | 4bits | **71.3** | **89.9** | **71.1** | **90.0** |
| PACT | 5bits | 71.1 | 89.6 | 71.2 | 89.8 |
| HAQ | flexible | 70.58 | 89.77 | 70.90 | 89.91 |
| SAT (Ours) | 5bits | **71.9** | **90.3** | **72.0** | **90.4** |
| PACT | 6bits | 71.2 | 89.2 | 71.5 | 90.0 |
| HAQ | flexible | 71.20 | 90.19 | 71.89 | 90.36 |
| SAT (Ours) | 6bits | **72.3** | **90.4** | **72.3** | **90.6** |
| PACT | 8bits | 71.3 | 89.7 | 71.7 | 89.9 |
| HAQ | flexible | 70.82 | 89.85 | 71.81 | 90.25 |
| SAT (Ours) | 8bits | **72.6** | **90.7** | **72.5** | **90.7** |
| PACT | FP | 72.1 | 90.2 | 72.1 | 90.5 |
| SAT (Ours) | FP | 71.7 | 90.2 | 71.8 | 90.2 |

Note that we give different result for the case of $x < \alpha$ in Eq. (13) compared to the original PACT [Choi et al. (2018)], where quantization error is ignored and the derivative of $q$ is approximated to 0, i.e., $q_k\left(\frac{\tilde{x}}{\alpha}\right) \approx \frac{\tilde{x}}{\alpha}$. Such quantization error will not introduce significant difference for high precisions, but the error introduced might be harmful for low precisions such as 4bit or lower. Notably, a concurrent work [Jain et al. (2019)] presents similar results.

To see the effect of this quantization error, we train MobileNet-V2 on ImageNet with low-precision of 2 and 4 bits, with the gradient calibrated as in Eq. (13) or not as in original PACT [Choi et al. (2018)]. The learning curves are illustrated in Fig. 3, which show that calibration of PACT gradient facilitates efficient training for low precision models. This gradient calibration technique is named CG-PACT for brevity.

## 4 EXPERIMENTS

Based on previous analysis, we apply the SAT and CG-PACT technqiues to popular models including MobileNet-V1, MobileNet-V2, PreResNet-50 on ImageNet. For all experiments, we use cosine learing rate scheduler [Loshchilov & Hutter (2016)] without restart. Learning rate is initially set to 0.05 and updated every iteration for totally 150 epochs. We use SGD optimizer, Nesterov momentum with a momentum weight of 0.9 without damping, and weight decay of $4 \times 10^{-5}$. The batch size

Table 2: Comparison of quantization techniques with only weights quantized.

| Quant. Method | Weights | MobileNet-V1 | | MobileNet-V2 | |
|---|---|---|---|---|---|
| | | Acc.-1 | Acc.-5 | Acc.-1 | Acc.-5 |
| Deep Compression | 2bits | 37.62 | 64.31 | 58.07 | 81.24 |
| HAQ | flexible | 57.14 | 81.87 | **66.75** | **87.32** |
| SAT (Ours) | 2bits | **66.3** | **86.8** | **66.8** | 87.2 |
| Deep Compression | 3bits | 65.93 | 86.85 | 68.00 | 87.96 |
| HAQ | flexible | 67.66 | 88.21 | 70.90 | 89.76 |
| SAT (Ours) | 3bits | **70.7** | **89.5** | **71.1** | **89.9** |
| Deep Compression | 4bits | 71.14 | 89.84 | 71.24 | 89.93 |
| HAQ | flexible | 71.74 | **90.36** | 71.47 | 90.23 |
| SAT (Ours) | 4bits | **72.1** | 90.2 | **72.1** | **90.6** |
| Deep Compression | FP | 70.90 | 89.90 | 71.87 | 90.32 |
| HAQ | FP | 70.90 | 89.90 | 71.87 | 90.32 |
| SAT (Ours) | FP | 71.7 | 90.2 | 71.8 | 90.2 |

Table 3: Comparison of quantization techniques on ResNet-50.

| Both Quantization | | | | Weight-Only Quantization | | | |
|---|---|---|---|---|---|---|---|
| Quant. Method[†] | Bit-widths | Acc.-1 | Acc.-5 | Quant. Method[†] | Weights | Acc.-1 | Acc.-5 |
| PACT | 2bits | 72.2 | 90.5 | DeepCompression | 2bits | 68.95 | 88.68 |
| LQNet | 2bits | 71.5 | 90.3 | LQNet | 2bits | 75.1 | 92.3 |
| LSQ | 2bits | 73.7 | 91.5 | HAQ | flexible | 70.63 | 89.93 |
| SAT (Ours) | 2bits | **74.1** | **91.7** | SAT (Ours) | 2bits | **75.3** | **92.4** |
| PACT | 3bits | 75.3 | 92.6 | DeepCompression | 3bits | 75.10 | 92.33 |
| LQNet | 3bits | 74.2 | 91.6 | LQNet | 3bits | NA | NA |
| LSQ | 3bits | 75.8 | 92.7 | HAQ | flexible | 75.30 | 92.45 |
| SAT (Ours) | 3bits | **76.6** | **93.1** | SAT (Ours) | 3bits | **76.3** | **93.0** |
| PACT | 4bits | 76.5 | 93.2 | DeepCompression | 4bits | 76.15 | 92.88 |
| LQNet | 4bits | 75.1 | 92.4 | LQNet | 4bits | **76.4** | **93.1** |
| LSQ | 4bits | 76.7 | 93.2 | HAQ | flexible | 76.14 | 92.89 |
| SAT (Ours) | 4bits | **76.9** | **93.3** | SAT (Ours) | 4bits | **76.4** | 93.0 |
| PACT | FP | 76.9 | 93.1 | DeepCompression | FP | 76.15 | 92.86 |
| LQNet | FP | 76.4 | 93.2 | LQNet | FP | 76.4 | 93.2 |
| LSQ | FP | 76.9 | 93.4 | HAQ | FP | 76.15 | 92.86 |
| SAT (Ours) | FP | 75.9 | 92.5 | SAT (Ours) | FP | 75.9 | 92.5 |

[*] PACT and SAT use full pre-activation ResNet, LSQ and HAQ use vanilla ResNet, and LQNet uses vanilla ResNet without convolution operation in shortcut (type-A shortcut).

[†] PACT and LQNet use full-precision for the first and last layers, LSQ and SAT use 8bit for both layers, and HAQ uses 8bit for the first layer.

is set to 2048 for MobileNet-V1/V2 and 1024 for PreResNet-50, and we adopt the warmup strategy suggested in [Goyal et al. (2017)] by linearly increasing the learning rate every iteration to a larger value ($\mathrm{batch\ size}/256 \times 0.05$) for the first five epochs before using the cosine annealing scheduler. The input image is randomly cropped to $224 \times 224$ and randomly flipped horizontally, and is kept as 8 bit unsigned integer with no standardization applied. Note that we use full-precision models with clamped weight as initial points to finetune quantized models.

We compare our method with techniques in recent literature, including uniform-precision quantization algorithms such as DeepCompression [Han et al. (2015)], PACT [Choi et al. (2018)], LQNet [Zhang et al. (2018)], LSQ [Esser et al. (2019)], and mixed-precision approaches such as

HAQ [Wang et al. (2019)] and AutoQB [Lou et al. (2019)]. Validation accuracy with respect to quantization levels for ResNet-50 is plotted in Fig. 1. It is obvious that our method gives significant and consistent improvement over previous methods under the same resource constraint. More thorough comparisons for quantization on MobileNets with or without quantized activation are given in Table 1 and 2, respectively. Table 3 compares different quantization techniques on ResNet-50. Surprisingly, the quantized models with our approach not only outperform all previous methods, including mixed-precision algorithms, but even outperform full-precision ones when the quantization is moderate ( $\geq 5$ bits for both quantization and 4 bits for weight-only quantization on MobileNet-V1/V2, $\geq 3$ bits for either both and weight-only quantization on ResNet-50).

Our method reveals that previous quantization techniques indeed suffer from inefficient training. With proper adjustment to abide by the efficient training rules, the quantized models achieve comparable or even better performance than their full-precision counterparts. In this case, we have to rethink about the doctrine in the model quantization literature that quantization itself hampers the capacity of the model. It seems with mild quantization, the generated models do not sacrifice in capacity, but benefit from the quantization procedure. The claimping and rescaling technique does not contribute to the gain in quantized models since they are already used in full-precision training. One potential reason is that quantization acts as a favorable regularization during training and help the model to generalize better. Note that quantizing both weight and activation gives better results than weight-only quantization in some cases (3 and 4 bits on ResNet-50), which also indicates that the quantization to the activations acts as a proper regularization and improves generalization capability of the network. The underlying mechanism is not clear yet. We left in-depth exploration as future work.

**Ablation Study** To see the different impacts of the two methods we proposed (SAT and CG-PACT), we experiment on MobileNet-V1 under two different quantization levels of 4bit and 8bit with combinations of the two approaches. As shown in Table 4, both techniques contribute to the performance, and SAT is more critical than CG-PACT in both settings. In mild quantization such as 8 bit, SAT itself is sufficient to achieve high performance, which matches our previous analysis.

Table 4: Ablation study of the SAT and CG-PACT on quantized MobileNet-V1. Here CG denotes CG-PACT method.

| | 4bits | | 8bits | |
|---|---|---|---|---|
| Quant. Method | Acc.-1 | Acc.-5 | Acc.-1 | Acc.-5 |
| PACT | 70.3 | 89.2 | 71.3 | 89.7 |
| PACT+SAT | 70.9 | 89.5 | **72.6** | 90.6 |
| PACT+CG | 70.4 | 89.3 | 71.0 | 89.5 |
| PACT+SAT+CG | **71.3** | **89.9** | **72.6** | **90.7** |

## 5 CONCLUSION

This paper studies efficient training of quantized neural networks. By analyzing the optimization of cross entropy loss and training dynamics in a neural network, we present a law describing the gradient flow during training with semi-quantitative analysis and empirical verification. Based on this, we suggest two rules of efficient training, and propose a scale-adjusted training technique to facilitate efficient training of quantized neural network. Our method yields state-of-the-art performance on quantized neural network. The efficient training rules not only applies to the quantization scenario, but can also be used as a inspection tool in other deep learning tasks.

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

# A SEMI-QUANTITATIVE ANALYSIS AND EMPIRICAL VERIFICATION OF LOGIT SCALE AND GRADIENT FLOW

In this section, we first make some semi-quantitative analysis of the scale of logit values and the gradient flow to derive the laws and rules introduced in Section 3.2. After that, we present some experimental results to verify the laws, together with some comments on our basic assumption. We want to emphasize that our analysis is only semi-quantitative to give insights for practice, and is not guaranteed to be rigorous. Complete and detailed analysis is far beyond the scope of this paper. In contrast, we will focus on order analysis of quantities, instead of accurate values, so we assume all variables are random unless specifically mentioned, and variables in the same tensor (weight matrix or activation matrix in a layer) are distributed identically. We will also widely adopt the assumption of independence between different random variables (with different names, layer indices or component indices), as claimed explicitly in the Axiom 3.2 of [Yang & Schoenholz (2017)]. This assumption is also adopted in other literature, such as [He et al. (2015); Schoenholz et al. (2016)]. It is not rigorously correct, but it leads to useful conclusion in practice [Xiao et al. (2018)]. In spite of this, we will give some comments on this assumption in the end of this section.

## A.1 SEMI-QUANTITATIVE ANALYSIS

As mentioned in Section 3.2, we discuss a simplified network composed of repeated blocks, each consisting of a linear (either convolution or fully-connected) layer without bias, a BN layer, the activation function and a pooling layer. The final stage is a single fully-connected layer without bias, which produces the logit values for the loss function. The pooling operation can be either average pooling or max pooling, but we only analyze the case of average pooling, and leave the analysis of max pooling as future work. We will first analyze the scale of logit values produced by the last fully-connected layer, and then discuss the training dynamics of the network.

### A.1.1 ANALYSIS OF THE LOGIT SCALE

The logit values produced by the final fully-connected layer is

$$z_i^{(L)} = \sum_j \Xi_{ij}^{(L)} x_j^{(L)} \tag{14}$$

where $L$ is the depth of the network, $\Xi^{(L)}$ is the effective weight of this layer, and $x^{(L)}$ is its input given by the previous block. The BN, nonlinear activation and pooling operations can be represented by

$$y_i^{(L-1)} = \gamma_i^{(L-1)} \frac{z_i^{(L-1)} - \mu_i^{(L-1)}}{\sigma_i^{(L-1)}} + \beta_i^{(L-1)} \tag{15a}$$

$$\widetilde{x}_i^{(L)} = h(y_i^{(L-1)}) \tag{15b}$$

$$x_i^{(L)} = \frac{1}{(k_{L-1}^{\text{pool}})^2} \sum_{p \in \mathcal{P}} \widetilde{x}_p^{(L)} \tag{15c}$$

From this, we have

$$\mathbb{VAR}[z_i^{(L)}] = n_L \mathbb{VAR}[\Xi_{ij}^{(L)}] \mathbb{E}[(x_j^{(L)})^2] \tag{16a}$$

$$= n_L \mathbb{VAR}[\Xi_{ij}^{(L)}] \frac{1}{(k_{L-1}^{\text{pool}})^4} \sum_p \mathbb{E}[(\widetilde{x}_p^{(L)})^2] \tag{16b}$$

$$= n_L \mathbb{VAR}[\Xi_{ij}^{(L)}] \frac{1}{(k_{L-1}^{\text{pool}})^2} \mathbb{E}[(\widetilde{x}_p^{(L)})^2] \tag{16c}$$

$$\approx n_L \mathbb{VAR}[\Xi_{ij}^{(L)}] \frac{1}{(k_{L-1}^{\text{pool}})^2} \mathbb{VAR}[y_i^{(L-1)}] \tag{16d}$$

$$\approx n_L \mathbb{VAR}[\Xi_{ij}^{(L)}] \frac{1}{(k_{L-1}^{\text{pool}})^2} (\gamma_i^{(L-1)})^2 \tag{16e}$$

where $n_L$ is the length of the input features of this last fully-connected layer. For a wide range of models including ResNets and MobileNets, $n_L$ equals to the number of output channels of the last convolution layer. Note that we have extensively used the assumption of independence, and rely on the fact that the activation function $h(\cdot)$ is quasi-linear.

Typically, the weight of the batch normalization layer $\gamma$ is on the order of $\mathcal{O}(1)$, and thus we have

$$\mathbb{VAR}[z_i^{(L)}] \approx n_L \mathbb{VAR}[\Xi_{ij}^{(L)}] \frac{1}{(k_{L-1}^{\text{pool}})^2} \tag{17}$$

To avoid output saturation, the softmax function for calculating the cross entropy loss has a preferred range of the logit values for efficient optimization. To get a more intuitive understanding of such preference, we plot the sigmoid function and its derivative in Fig. 4 as a simplification of softmax function. It is clear that for efficient optimization, the scale of the input $x$ should be well below 1, from which we can derive that

$$\mathbb{VAR}[\Xi_{ij}^{(L)}] \ll \frac{(k_{L-1}^{\text{pool}})^2}{n_L} \tag{18}$$

which is the Efficient Training Rule I.

For more intuitive understanding of the ETR I, here we analyze the impact of clamping operation for some typical models, based on the variance ratio given in Fig. 2a. The results are illustrated in Table 5, where the variance of the original weight $\mathbb{VAR}[W_{ij}^{(L)}]$ is given by

$$\mathbb{VAR}[W_{ij}^{(L)}] = \frac{1}{\widehat{n}_L} \tag{19}$$

and we need to determine if the value of $\kappa_0$ defined as

$$\kappa_0 := \frac{1}{(k_{L-1}^{\text{pool}})^2} \cdot n_L \mathbb{VAR}[\Xi_{ij}^{(L)}] \tag{20}$$

is sufficiently small in comparison with 1. Note that $\Xi = W$ for vanilla models and $\Xi = \widehat{W}$ for models with clamping. From the results we can see that models with clamping have much larger $\kappa_0$, which results in violation of the ETR I. This explains the degeneration of the learning curve for clamping in Fig. 2b.

Table 5: Impact of clamping on ResNet-18/50 and MobileNet-V1/V2 with ImageNet.

| model | ResNet-18 | ResNet-50 | MobileNet-V1 | MobileNet-V2 |
|---|---|---|---|---|
| $n_L$ | 512 | 512×4 | 1024 | 1280 |
| $\widehat{n}_L = n_{L+1}$ | 1000 | 1000 | 1000 | 1000 |
| $k_{L-1}^{\text{pool}}$ | 7 | 7 | 7 | 7 |
| $\kappa_0^{\text{vanilla}}$ | 0.01 | 0.04 | 0.02 | 0.026 |
| $\kappa_0^{\text{clamp}}$ | 0.5 | 2 | 1 | 1.3 |

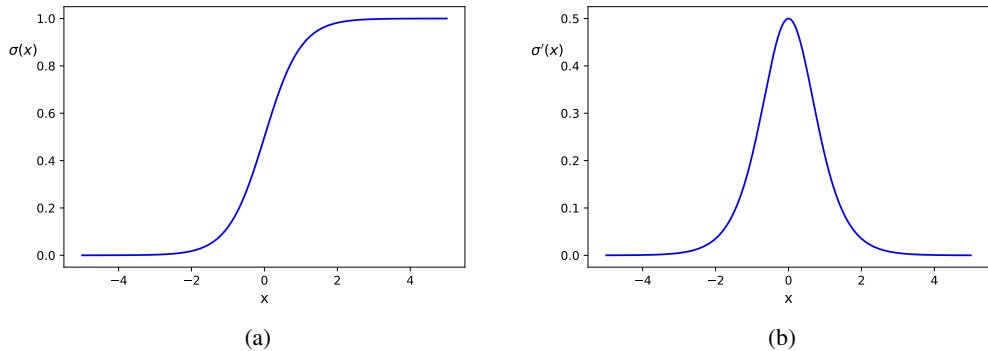

Figure 4: Plot of the sigmoid function (a) and its derivative (b).

### A.1.2 TRAINING DYNAMICS

Now we analyze the training dynamics of the network. Suppose for the $l$-th block, the input is $x^{(l)}$, and the effective weight of the linear layer is $\Xi^{(l)}$. In other words, we have

$$z_i^{(l)} = \sum_j \Xi_{ij}^{(l)} x_j^{(l)} \tag{21a}$$

$$y_i^{(l)} = \gamma_i^{(l)} \frac{z_i^{(l)} - \mu_i^{(l)}}{\sigma_i^{(l)}} + \beta_i^{(l)} \tag{21b}$$

$$\widetilde{x}_i^{(l+1)} = h(y_i^{(l)}) \tag{21c}$$

$$x_i^{(l+1)} = \frac{1}{(k_l^{\text{pool}})^2} \sum_{p \in \mathcal{P}} \widetilde{x}_p^{(l+1)} \tag{21d}$$

where $l = 1, \ldots, L - 1$.

Since the parameters $\gamma^{(l)}$ and $\beta^{(l)}$ are constants (although trainable), and the activation function is quasi-linear, we have

$$\mathbb{VAR}[y_i^{(l)}] \approx (\gamma^{(l)})^2 \tag{22a}$$

$$\mathbb{E}[(\widetilde{x}_i^{(l+1)})^2] \sim \mathbb{VAR}[y_i^{(l)}] \approx (\gamma^{(l)})^2 \tag{22b}$$

$$\mathbb{E}[(x_i^{(l+1)})^2] = \frac{1}{(k_l^{\text{pool}})^2} \mathbb{E}[(\widetilde{x}_i^{(l+1)})^2] \approx \left( \frac{\gamma^{(l)}}{k_l^{\text{pool}}} \right)^2 \tag{22c}$$

$$(\sigma_i^{(l)})^2 = \mathbb{VAR}[z_i^{(l)}] \tag{22d}$$

$$= n_l \mathbb{VAR}[\Xi_{ij}^{(l)}] \mathbb{E}[(x_j^{(l)})^2] \tag{22e}$$

$$\approx n_l \mathbb{VAR}[\Xi_{ij}^{(l)}] \left( \frac{\gamma^{(l-1)}}{k_{l-1}^{\text{pool}}} \right)^2 \tag{22f}$$

where $n_l$ is the number of input neurons in the $l$-th layer, i.e. $n_l = c_l \cdot (k_l^{\text{conv}})^2$, with $c_l$ and $k_l^{\text{conv}}$ denoting the number of input channels and kernel size of the convolution layer, respectively. For fully-connnected layer, we can set $k_l^{\text{conv}} = 1$. Here, we have omitted the indices of trainable parameters in the batch normalization layer by simply assuming that they are of the similar order. The second equality holds because the activation function is quasi-linear, and in the third equality we have used the assumption of independence. Meanwhile, in the last three equalities, we have used the definition of $\sigma^{(l)}$ and also the assumption that $W^{(l)}$ and $x^{(l)}$ are all independent with each other.

During training, both $\mu_i^{(l)}$ and $\sigma_i^{(l)}$ are functions of $z_i^{(l)}$, and we have

$$\frac{\partial \mu_i^{(l)}}{\partial z_i^{(l)}} = \frac{1}{m_B} \tag{23a}$$

$$\frac{\partial \sigma_i^{(l)}}{\partial z_i^{(l)}} = \frac{2}{m_B} \frac{z_i^{(l)} - \mu_i^{(l)}}{\sigma_i^{(l)}} \tag{23b}$$

therefore,

$$
\begin{aligned}
\frac{\partial y_i^{(l)}}{\partial z_i^{(l)}} &= \gamma^{(l)} \left[ \frac{1}{\sigma_i^{(l)}} - \frac{1}{\sigma_i^{(l)}} \frac{\partial \mu_i^{(l)}}{\partial z_i^{(l)}} + (z_i^{(l)} - \mu_i^{(l)}) \frac{-1}{(\sigma_i^{(l)})^2} \frac{\partial \sigma_i^{(l)}}{\partial z_i^{(l)}} \right] \\
&= \gamma^{(l)} \left[ \frac{1}{\sigma_i^{(l)}} - \frac{1}{\sigma_i^{(l)}} \frac{1}{m_B} - \frac{1}{\sigma_i^{(l)}} \frac{2}{m_B} \frac{(z_i^{(l)} - \mu_i^{(l)})^2}{(\sigma_i^{(l)})^2} \right] \\
&\approx \frac{\gamma^{(l)}}{\sigma_i^{(l)}} \quad (m_B \gg 1)
\end{aligned}
\tag{24}
$$

Here, $m_B$ is the batch size.

From the above results, we can derive the relationship between the gradients with respect to the weights and activations in two adjacent layers. Actually, from Eq. (21) and Eq. (24), we have

$$\partial_{\Xi_{ij}^{(l)}} \mathcal{L} \approx x_j^{(l)} \frac{\gamma^{(l)}}{\sigma_i^{(l)}} h'(y_i^{(l)}) \partial_{\widetilde{x}_i^{(l+1)}} \mathcal{L} \tag{25a}$$

$$\partial_{x_j^{(l)}} \mathcal{L} = \sum_i \Xi_{ij}^{(l)} \frac{\gamma^{(l)}}{\sigma_i^{(l)}} h'(y_i^{(l)}) \partial_{\widetilde{x}_i^{(l+1)}} \mathcal{L} \tag{25b}$$

$$\partial_{\widetilde{x}_p^{(l+1)}} \mathcal{L} = \frac{1}{(k_l^{\text{pool}})^2} \partial_{x_i^{(l+1)}} \mathcal{L} \tag{25c}$$

The variances of the gradients are thus given by

$$\mathbb{VAR}[\partial_{\Xi_{ij}^{(l)}} \mathcal{L}] = \left( \frac{\gamma^{(l)}}{\sigma_i^{(l)}} \right)^2 \mathbb{E}[(x_j^{(l)})^2] \mathbb{E}[(h'(y_i^{(l)}))^2] \mathbb{VAR}[\partial_{\widetilde{x}_i^{(l+1)}} \mathcal{L}] \tag{26a}$$

$$\mathbb{VAR}[\partial_{x_j^{(l)}} \mathcal{L}] = \widehat{n}_l \left( \frac{\gamma^{(l)}}{\sigma_i^{(l)}} \right)^2 \mathbb{VAR}[\Xi_{ij}^{(l)}] \mathbb{E}[(h'(y_i^{(l)}))^2] \mathbb{VAR}[\partial_{\widetilde{x}_i^{(l+1)}} \mathcal{L}] \tag{26b}$$

$$\mathbb{VAR}[\partial_{\widetilde{x}_p^{(l+1)}} \mathcal{L}] = \frac{1}{(k_l^{\text{pool}})^4} \mathbb{VAR}[\partial_{x_i^{(l+1)}} \mathcal{L}] \tag{26c}$$

where $\widehat{n}_l = c_{l+1} \cdot (k_l^{\text{conv}})^2$ and $c_{l+1}$ is the number of output channels. We use the assumption of independence for different variables here again.

From this we can get

$$\mathbb{VAR}[\partial_{\Xi_{ij}^{(l)}} \mathcal{L}] = \left( \frac{\gamma^{(l)}}{\sigma_i^{(l)}} \right)^2 \frac{\mathbb{E}[(x_j^{(l)})^2]}{\mathbb{E}[(x_m^{(l+1)})^2]} \cdot \frac{\mathbb{E}[(h'(y_i^{(l)}))^2]}{(k_l^{\text{pool}})^4} \cdot \widehat{n}_{l+1} \mathbb{VAR}[\Xi_{ki}^{(l+1)}] \mathbb{VAR}[\partial_{\Xi_{km}^{(l+1)}} \mathcal{L}] \tag{27}$$

Note that the subscripts of variables inside $\mathbb{VAR}$ can be eliminated as variables in the same tensor are distributed identically by assumption.

If we have batch normalization layer after the linear layer, based on Eq. (22c) and Eq. (22e), we have

$$\mathbb{VAR}[\partial_{\Xi_{ij}^{(l)}} \mathcal{L}] = \kappa_1^{(l)} \cdot \frac{1}{(k_l^{\text{pool}})^2} \cdot \frac{\widehat{n}_{l+1} \mathbb{VAR}[\Xi_{ij}^{(l+1)}]}{n_l \mathbb{VAR}[\Xi_{ij}^{(l)}]} \mathbb{VAR}[\partial_{\Xi_{ij}^{(l+1)}} \mathcal{L}] \tag{28}$$

Here, $\kappa_1^{(l)}$ is some empirical parameter of order $\mathcal{O}(1)$ with respect to $n_l$.

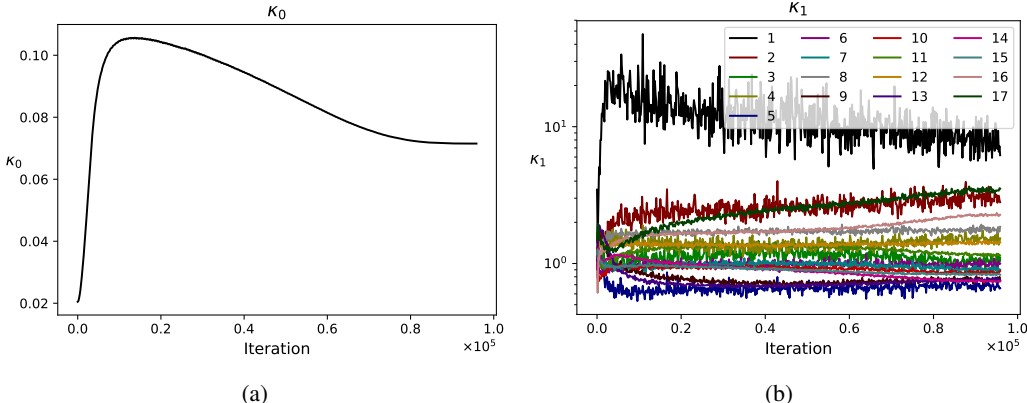

Figure 5: Empirical verification of Eq. (18) and Eq. (28). (a) $\kappa_0$ calculated with Eq. (30a). (b) $\kappa_1$ given by Eq. (30b) for different layers.

On the other hand, if there is no batch normalization layers, $\gamma^{(l)}$ and $\sigma_i^{(l)}$ will be removed from Eq. (27) and $\mathbb{VAR}[\partial_{\Xi_{ij}^{(l)}}\mathcal{L}]$ can be formulated as

$$\mathbb{VAR}[\partial_{\Xi_{ij}^{(l)}}\mathcal{L}] = \kappa_2^{(l)} \cdot \frac{1}{(k_l^{\mathrm{pool}})^4} \cdot \widehat{n}_{l+1}\mathbb{VAR}[\Xi_{ij}^{(l+1)}]\mathbb{VAR}[\partial_{\Xi_{ij}^{(l+1)}}\mathcal{L}] \tag{29}$$

For typical case, $\kappa_2^{(l)}$ is another empirical parameter of order $\mathcal{O}(1)$ with respect to $n_l$.

Both Eq. (28) and Eq. (29) are referred to as the gradient flowing law.

## A.2 EMPIRICAL VERIFICATION

We are now ready to verify Eq. (18) and Eq. (28) with experiments. For this purpose, we train a vanilla ResNet18 with Kaiming-initialization [He et al. (2015)] and batch normalization to guarantee convergent training. In this case, we have $\Xi = W$. We calculate the two parameters $\kappa_0$ and $\kappa_1$ given by

$$\kappa_0 := \frac{1}{(k_{L-1}^{\mathrm{pool}})^2} \cdot n_L \mathbb{VAR}[W_{ij}^{(L)}] \tag{30a}$$

$$\kappa_1^{(l)} := (k_l^{\mathrm{pool}})^2 \cdot \frac{n_l \mathbb{VAR}[W_{ij}^{(l)}]}{\widehat{n}_{l+1}\mathbb{VAR}[W_{ij}^{(l+1)}]} \cdot \frac{\mathbb{VAR}[\partial_{W_{ij}^{(l)}}\mathcal{L}]}{\mathbb{VAR}[\partial_{W_{ij}^{(l+1)}}\mathcal{L}]} \tag{30b}$$

Here $\kappa_0$ is calculated using statistics from the last fully-connected layer and $\kappa_1$ is calculated for all convolution layers along the whole training procedure. We plot the values of $\kappa_0$ and $\kappa_1$ in Fig. 5. As we can see, $\kappa_0 \ll 1$ during the whole training process as expected. $\kappa_1^{(l)}$ in different layers are all of order $\mathcal{O}(1)$ during the training process, except for the first layer, which involves max pooling and its analysis is out of our scope (we set $k_1^{\mathrm{pool}} = 1$ in the computation for this layer). Nevertheless, we notice that the discrepancy is not large.

## A.3 COMMENTS ON THE INDEPENDENCE ASSUMPTION

Here we want to present some further discussion on the independence assumption, which is extensively used in our analysis. Our discussion is to facilitate more intuitive understanding of such assumption, but not in pursue of rigorous proof.

The first point we want to emphasize is related to the central limit theorem (CLT). Independence is one important condition for the CLT, but a sufficiently weak dependence does not harm [Van Kampen (1992)]. Typical examples where independence does not hold include the momentum of molecules in an ideal gas described by the micro-canonical ensemble, or random walk with persistence. Even with independence assumption violated, the sum of random variables is able to

converge to a Gaussian distribution as the number of random variables increases indefinitely. Thus, with sufficient number of neurons, linear operation can result in normally distributed outputs, which further leads to the normal distribution of gradients. Now since the weights are initialized with normal distribution, updating with normally distributed gradients will result in some new Gaussian random variables, and thus we can focus on the variance to characterize their statistical properties. Experiment shows that the trained parameters still resemble Gaussian distribution (sometimes with some small spurs), especially for the last several layers.

Second, we investigate a simple example to show how the independence property can be derived with zeroth-order approximation. For simplicity, we analyze the product of two random variables $U$ and $V$, both of which are some function of another random variable $X$, that is

$$U = u(X) \tag{31a}$$
$$V = v(X) \tag{31b}$$

We can see that $U$ and $V$ are generally not independent.

The random variables can be viewed as a quantity with some true value given by the mean, but perturbed by some small random noise, that is

$$U = \mathbb{E}[U] + \delta U \tag{32a}$$
$$V = \mathbb{E}[V] + \delta V \tag{32b}$$

and we have

$$\mathbb{E}[UV] = \mathbb{E}[(\mathbb{E}[U] + \delta U)(\mathbb{E}[V] + \delta V)] \tag{33a}$$
$$= \mathbb{E}[U]\mathbb{E}[V] + \mathbb{E}[U]\mathbb{E}[\delta V] + \mathbb{E}[\delta U]\mathbb{E}[V] + \mathbb{E}[\delta U \delta V] \tag{33b}$$
$$= \mathbb{E}[U]\mathbb{E}[V] + \mathbb{E}[\delta U \delta V] \tag{33c}$$

For zeroth-order approximation, we can ignore the second term, which is a second-order quantity, and get

$$\mathbb{E}[UV] \approx \mathbb{E}[U]\mathbb{E}[V] \tag{34}$$

For variance of $UV$, if $\mathbb{E}[U] = 0$, we have

$$\mathbb{VAR}[UV] = \mathbb{E}[U^2 V^2] - (\mathbb{E}[UV])^2 \tag{35a}$$
$$\approx \mathbb{E}[U^2 V^2] \tag{35b}$$
$$\approx \mathbb{E}[U^2]\mathbb{E}[V^2] \tag{35c}$$
$$= \mathbb{VAR}[U]\mathbb{E}[V^2] \tag{35d}$$

which is an approximated independence property widely used in our derivation. Note that such zeroth-order approximation is broadly adopted in statistical mechanics and related areas for analyzing macroscopic law of noisy physical systems, where nonlinear mechanisms exist [Van Kampen (1992)]. More accurate results could be derived by expansion of the master equation, which is beyond the scope of this paper. Such zeroth-order approximation is also similar to the annealed approximation method in condensed matter physics [Seung et al. (1992)].

## B  IMPACT OF QUANTIZATION ON WEIGHT VARIANCE

In this section, we examine the impact of quantization on weight variance. Fig. 6 shows the ratio between the standard deviations of them with respect to the number of bits for different channel numbers, which determines the variances of the original non-clamped weights $W$. We can find that for precision higher than 3 bits, quantized weights have nearly the same variance as the full-precision weights, indicating quantization itself introduces little impact on training. However, the discrepancy increases significantly for low precision such as 1 or 2 bits. Also, different channel numbers give similar results.

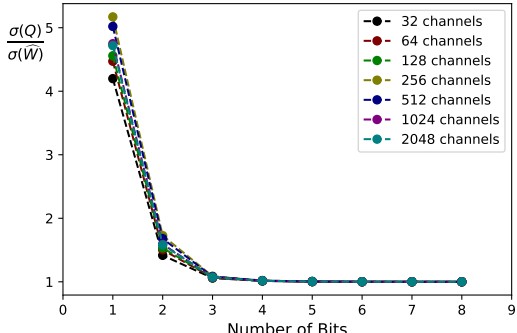

Figure 6: Impact of weight quantization on the variance of effective weight under different channel numbers.

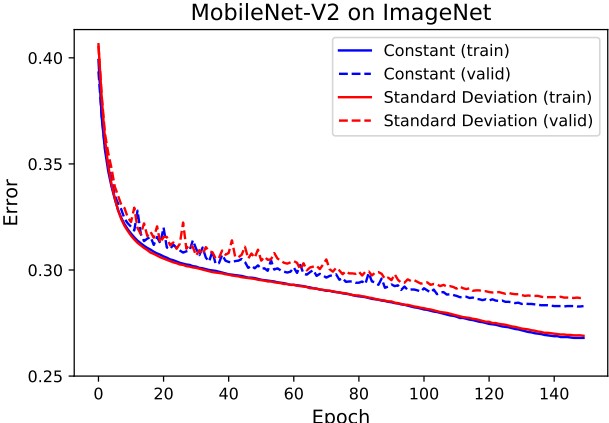

Figure 7: Comparison of constant rescaling and rescaling with standard deviation.

## C  COMPARISON OF RESCALING METHOD

In this section, we will compare the constant rescaling methods proposed in 3.3 and another method named rescaling with standard deviation. It standardizes the effective weights and then rescales them with the standard deviation of the original weights as in Eq. (36), where $\mathbb{VAR}[\cdot]$ is the sample variance of elements in the weight matrix, calculated by averaging the square of elements in the weight matrix.

$$W_{ij}^{*} = \sqrt{\frac{\mathbb{VAR}[W_{rs}]}{\mathbb{VAR}[\widehat{W}_{rs}]}} \, \widehat{W}_{ij} \tag{36}$$

We train MobileNet-V2 on ImageNet with weight clamping and rescale the last fully-connected layer using the two methods, and plot their learning curves as shown in Fig. 7. We find that the two methods give similar results.

## D    GRADIENT CALCULATION IN PACT

Here we derive the result in Eq. (13). The activation value $x$ is clamped into the interval $[0, \alpha]$ and then quantized as follows

$$\widetilde{x} = \frac{1}{2}\Big[|x| - |x - \alpha| + \alpha\Big] \tag{37a}$$

$$= \begin{cases} 0 & x < 0 \\ x & 0 < x < \alpha \\ \alpha & x > \alpha \end{cases} \tag{37b}$$

$$q = \alpha q_k\left(\frac{\widetilde{x}}{\alpha}\right) \tag{37c}$$

Based on the STE rule, we have

$$q'_k(x) \coloneqq 1 \tag{38}$$

Taking derivative with respect to $\alpha$ in Eq. (37), we have

$$\frac{\partial q}{\partial \alpha} = q_k\left(\frac{\widetilde{x}}{\alpha}\right) + \alpha\frac{\partial}{\partial \alpha}q_k\left(\frac{\widetilde{x}}{\alpha}\right) \tag{39a}$$

$$= q_k\left(\frac{\widetilde{x}}{\alpha}\right) + \alpha \cdot 1 \cdot \frac{\partial}{\partial \alpha}\left(\frac{\widetilde{x}}{\alpha}\right) \tag{39b}$$

$$= q_k\left(\frac{\widetilde{x}}{\alpha}\right) + \alpha \cdot \frac{1}{\alpha}\frac{\partial \widetilde{x}}{\partial \alpha} + \alpha\widetilde{x}\left(-\frac{1}{\alpha^2}\right) \tag{39c}$$

$$= \begin{cases} q_k\left(\frac{\widetilde{x}}{\alpha}\right) + \alpha \cdot \frac{1}{\alpha} \cdot 0 + \alpha\widetilde{x}\left(-\frac{1}{\alpha^2}\right) & x < \alpha \\ q_k\left(\frac{\widetilde{x}}{\alpha}\right) + \alpha \cdot \frac{1}{\alpha} \cdot 1 + \alpha\widetilde{x}\left(-\frac{1}{\alpha^2}\right) & x > \alpha \end{cases} \tag{39d}$$

$$= \begin{cases} q_k\left(\frac{\widetilde{x}}{\alpha}\right) - \frac{\widetilde{x}}{\alpha} & x < \alpha \\ 1 + \alpha \cdot \frac{1}{\alpha} \cdot 1 + \alpha \cdot \alpha \cdot \left(-\frac{1}{\alpha^2}\right) & x > \alpha \end{cases} \tag{39e}$$

$$= \begin{cases} q_k\left(\frac{\widetilde{x}}{\alpha}\right) - \frac{\widetilde{x}}{\alpha} & x < \alpha \\ 1 & x > \alpha \end{cases} \tag{39f}$$

which is the result in Eq. (13).

## E    BITWIDTHS OF THE FIRST AND LAST LAYERS

Here we study the impact of quantization levels of the first and the last layers. Using MobileNet-V1, we compare the two settings of quantizing these two layers to a fixed 8 bits or to the same bit-width as other layers. As shown in Table 6, we find that the accuracy reduction is negligible for quantization levels higher than 4bits.

Table 6: Impact of precisions of the first and the last layers on MobileNet-V1.

| Bitwidths of Internal Layers | 8bits Both Layers | | Uniform Quantization | |
|:---:|:---:|:---:|:---:|:---:|
| | Acc.-1 | Acc.-5 | Acc.-1 | Acc.-5 |
| 4bits | 71.3 | 89.9 | 71.2 | 89.8 |
| 5bits | 71.9 | 90.3 | 72.1 | 90.3 |
| 6bits | 72.3 | 90.4 | 72.5 | 90.5 |

## F    RESCALING CONVOLUTION LAYERS WITHOUT BATCH NORMALIZATION FOLLOWED

In this section, we study the impact of rescaling to convolution layers without BN layers followed. As shown in Fig. 8, we plot the learning curves of Pre-ResNet50 with 2 bits weight-only quantization

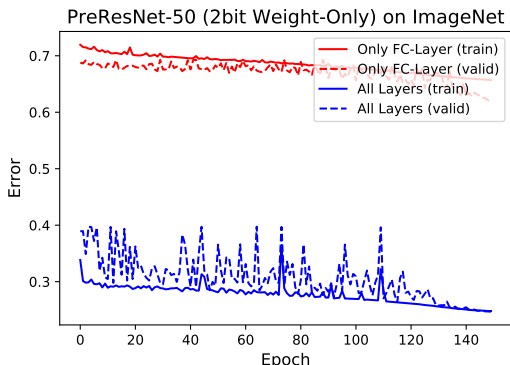

Figure 8: Learning curves for PreResNet-50 with 2bit weight-only quantization on ImageNet. For red lines, rescaling is only applied to the last fully-connected layer, while for blue lines, all layers are rescaled.

on ImageNet, with constant rescaling applied to only the fully-connected layer or all layers. We can see that applying rescaling to only the last fully-connected layer is not sufficient for efficient training, and rescaling on all convolution layers successfully achieves efficient training.

## G  ELIMINATION OF BATCH NORMALIZATION LAYERS

As mentioned in Section 3.1, batch normalization layers are not quantized in our experiments, and full-precision multiplication involved in BN is a considerable amount of computation during inference. Here, we want to discuss the possibility to eliminate such layers, especially for light-weight models such as MobileNet-V1/V2.

As shown in Eq. (8), quantization of activation function already involves full-precision multiplication, so we want to examine if it is possible to absorb the batch normalization operations into this clipping function. The quantized input to a linear layer can be written as

$$q = \frac{\alpha}{a} \cdot n \tag{40}$$

where $n$ is the integer output from the rounding function in quantizing the activation of previous layer, $a = 2^b - 1$ is determined by the bitwidth $b$ of the previous layer, and $\alpha$ is the clipping level of the activation function in the previous layer. The quantized output of the current layer is given by

$$\widetilde{q} = \frac{\widetilde{\alpha}}{\widetilde{a}} \left\lfloor \frac{\widetilde{a}}{\widetilde{\alpha}} \cdot \text{clipping}\Big(\gamma_i (Qq)_i + \beta_i, 0, \widetilde{\alpha}\Big) \right\rceil \tag{41a}$$

$$= \frac{\widetilde{\alpha}}{\widetilde{a}} \left\lfloor \frac{\widetilde{a}}{\widetilde{\alpha}} \gamma_i \cdot \text{clipping}\Big((Qq)_i + \frac{\beta_i}{\gamma_i}, 0, \frac{\widetilde{\alpha}}{\gamma_i}\Big) \right\rceil \tag{41b}$$

$$= \frac{\widetilde{\alpha}}{\widetilde{a}} \left\lfloor \frac{\widetilde{a}}{\widetilde{\alpha}} \gamma_i \cdot \text{clipping}\Big((\frac{\alpha}{a}Qn)_i + \frac{\beta_i}{\gamma_i}, 0, \frac{\widetilde{\alpha}}{\gamma_i}\Big) \right\rceil \tag{41c}$$

$$= \frac{\widetilde{\alpha}}{\widetilde{a}} \left\lfloor \frac{\widetilde{a}}{\widetilde{\alpha}} \frac{\alpha}{a} \gamma_i \cdot \text{clipping}\Big((Qn)_i + \frac{\beta_i}{\gamma_i}\frac{a}{\alpha}, 0, \frac{\widetilde{\alpha}}{\gamma_i}\frac{a}{\alpha}\Big) \right\rceil \tag{41d}$$

where $\widetilde{\alpha}$ and $\widetilde{a}$ are corresponding parameters for the current layer, $\text{clipping}(\cdot)$ is the clipping function given by Eq. (11a), and we have simplified the batch normalization by absorbing the running statistics into the parameters. Here, we have focused on one channel of the batch normalization layer, but the generalization is straightforward. From this, we can see that we only need to use different biasing and clipping levels for different channels, which can be pre-computed. The full-precision multiplications in BN layers are eliminated. Meanwhile, since $Q$ and $n$ are both quantized, lower precision can be adopted for computing the clipping levels.

Note that the above discussion is only applicable to models without skip-connection, such as MobileNet-V1, or models where convolution is not involved in skip-connection, such as MobileNet-V2. For ResNet and PreResNet, the above method is not applicable and more sophisticated methods might be necessary to eliminate BN layers.

