# OpenReview forum: "Rethinking Neural Network Quantization"
_ICLR.cc/2020/Conference — Reject_

### Official Review · AnonReviewer3 · 2019-10-23
**Official Blind Review #3**

**Rating:** 3

**Review:**

This paper presents network quantization techniques for weight quantization (SAT) and activation quantization (CG-PACT). The authors first formulate efficient training rules (ETR I and ETR II) and propose a normalization scheme for weight quantization based on the rules. Independently, the authors also propose an activation quantization rule by removing the approximated term in PACT. The authors present the effectiveness of their algorithm in MobileNet V1, V2, and PreResNet-50.

Overall, I think the proposed algorithms lack novelty due to the following reasons.

- SAT suggests a normalization scheme under some assumptions (ETR I and ETR 2), however, I am not sure this is a significant contribution compared to DoReFa. In particular, I have some doubt in assumptions which is listed in ‘Other comments’.

- CG-PACT seems to be a simple variant of PACT. In methodological contribution, I believe that it could only be a marginal improvement with respect to PACT. In terms of empirical contribution, it was hard to see the improvement made by CG-PACT as PACT and PACT+CG exhibit similar performance in Table 4.

Other comments
- In the derivation of ETR 1, some approximations are not clear for me. For example, the authors drop \gamma in (17). In addition, approximation in (16d) depends on the activation function and BN parameters but they are also ignored. I believe that incorporating all these would not introduce much computational overhead.

- In (3) and (4), I think that equality only holds under some assumptions (e.g., an infinite number of parameters, i.i.d. weights, etc.) but they are not clearly mentioned.


**Experience Assessment:**

I have read many papers in this area.

**Review Assessment: Checking Correctness Of Derivations And Theory:**

N/A

**Review Assessment: Checking Correctness Of Experiments:**

I assessed the sensibility of the experiments.

**Review Assessment: Thoroughness In Paper Reading:**

I read the paper at least twice and used my best judgement in assessing the paper.

---

> ### Author Response · Authors · 2019-11-07
> **Reply to Review #3**
>
> We appreciate the kind review from the reviewer. According to the comments, there might be some misunderstanding to our approach from the reviewer and we are glad to give more detailed explanations.
>
> In order to provide a semi-quantitative analysis on the key factors of model quantization, we do simplifications of the neural network parameters and also consider the most common configurations. For activation functions, since ReLU is the most generally adopted activation function, we only consider that (the same is applied in the paper https://arxiv.org/abs/1502.01852). For batch normalization weights, empirically it is on the order of O(1), so we can safely omit it without quite being misled. Analysis with details of \gamma and different activation functions will nevertheless be more complicated and we leave it as future work.
>
> For (3) and (4), the condition for the law is detailed in the appendix, it holds for most cases except some extreme cases (for general practical case, the approximation effect is incorporated in the parameter of \kappa’s).
>
> For CG-PACT, CG is not important for high-precision (for example, 8 bit), as the quantization error is very small. However, for low precision such as 4 bit, the difference can be as large as 0.4% for ImageNet classification, which is not negligible. Lower precision such as 2 bit will further exaggerate the problem as shown in Figure 3 in the paper.

---

### Official Review · AnonReviewer2 · 2019-10-23
**Official Blind Review #2**

**Rating:** 3

**Review:**

This paper proposes two rules for efficient training of quantized networks by investigating the scale of the logit values and gradient flow. The authors claim that accuracy degradation of recent quantization methods results from the violation of these two rules.

One of my main concerns is that the analysis of the rules for weight and activation quantization are separated. E.g., the analysis of weight quantization in Section 3.2 is based on eq (1a)-(1d) where no activation quantization is considered. In this case, does the analysis still hold when applying weight and activation are quantized simultaneously?

Moreover, the analysis is only suited for a limited range of quantization methods. In the proposed SAT, the authors propose to multiplies the normalized weight with the square root of the reciprocal of the number of neurons in the linear layer, to make up for the variance difference caused by quantization. However, this increase indeed depends on the initialization of the weights. If the weights are not sampled from "a Gaussian distribution of zero mean and variance proportional to the reciprocal of the number of neurons" as at the end of page 5, then this recipe may not work any longer. Moreover, the proposed SAT seems to be only suited for the specific quantization function for Dorefa-Net in (5), what about many other recent quantization functions that do not need this kind of clamping?

Others:
1. The citation format is wrong.
2. In the abstract, "Recent quantization approaches violates ... and results ..." => "Recent quantization approaches violate ... and result ..."
3. What is the "scaling factor in Eq. (3)" before the subsection "Efficient Training Rule II (ETR II)"?
4. Keep the same number of decimal places in the tables.

**Experience Assessment:**

I have published one or two papers in this area.

**Review Assessment: Checking Correctness Of Derivations And Theory:**

I assessed the sensibility of the derivations and theory.

**Review Assessment: Checking Correctness Of Experiments:**

I assessed the sensibility of the experiments.

**Review Assessment: Thoroughness In Paper Reading:**

I read the paper at least twice and used my best judgement in assessing the paper.

---

> ### Author Response · Authors · 2019-11-07
> **Reply to Review #2**
>
> We appreciate the kind review from the reviewer. According to the comments, There might be some misunderstanding to our approach from the reviewer and we are glad to give more detailed explanations.
>
> For the analysis in Section 3.2, it is quite generally applicable and there is no restriction on the value of any variables, and no matter they are full-precision or quantized, the analysis applies without any difference.
>
> For weight initialization, if there is no batch normalization layers followed, Kaiming initialization is the common strategy for correct convergence during training, as already analyzed in detail in the famous work (https://arxiv.org/abs/1502.01852). With batch normalization layers followed, initialization of convolution layers can be arbitrary, and we did not apply our techniques for these cases (as we mentioned, either batch normalization layer is applied, or the variance of effective weights should be proportional to the reciprocal of the number of neurons).
>
> DoReFa is a promising and widely adopted quantization strategy for model quantization, and is the basis for a bunch of modern quantization procedure such as PACT. This is the main reason we adopt this method in our work. Due to the similarity of the quantization procedure of DoReFa with other approaches such as XNORNet, Tenary Quant, HWGQ, etc, we believe our approach can be adopted to improve these algorithms as well.
>
> For the ‘scaling factor in Eq. (3)’ before the subsection “Efficient Training Rule II (ETR II)”, we mean the product in front of the variance of weight gradient on the right hand side of the equation.

---

### Official Review · AnonReviewer1 · 2019-11-02
**Official Blind Review #1**

**Rating:** 1

**Review:**

Quick Summary: Based on semi-quantitative analysis the paper first proposes two rules for quantization of DNNs, then extends previous methods based on these rules to propose specific technique for quantizing activations and weights. Experimental results are provided on MobileNet V1/VB2 and ResNet50 and compare favourably against baselines (which are old and unfortunately do not represent recent developments in the literature). Overall I found novelty low and also there are several recent papers already published before this submission which provide results which are at least as good or even better

Details
Specifically the rules are:
1. To prevent logits from entering saturation region of the cross entropy loss, the effective weight in the last fully connected layer should be small.
2.To keep the gradient of weights in the same scale across the whole network, either BN layers should be used after linear layers such as convolution and fully- connected layers, or the variance of the effective weights should be on the order of the reciprocal of the number of neurons of the linear layer (n_l).
They use DoReFa for weight quantization and PACT for activation quantization based on the above observations/rules. However the contributions were not really very novel in my opinion. See especially the paper "LEARNED STEP SIZE QUANTIZATION" I linked below when explaining novelty

This reviewer feels that the authors were perhaps unaware of several important contributions to the literature this year which are substantially better than the baselines compared against in the paper. I list a few below for the authors to compare against, and to explain their novelty in contributions in the rebuttal phase. Unfortunately this makes me feel the paper should be a clear reject in its current state, unless the authors can convince us otherwise.

https://arxiv.org/pdf/1902.08153.pdf
https://arxiv.org/pdf/1903.08066.pdf
https://arxiv.org/pdf/1905.11452.pdf
https://arxiv.org/pdf/1905.13082.pdf


**Experience Assessment:**

I have published one or two papers in this area.

**Review Assessment: Checking Correctness Of Derivations And Theory:**

I assessed the sensibility of the derivations and theory.

**Review Assessment: Checking Correctness Of Experiments:**

I assessed the sensibility of the experiments.

**Review Assessment: Thoroughness In Paper Reading:**

I read the paper thoroughly.

---

> ### Author Response · Authors · 2019-11-07
> **Reply to Review #1**
>
> We appreciate the kind review from the reviewer, but it seems a bit unfortunate that the reviewer lost some important insights we provided and did not check the results very carefully. As far as we know, the four papers the reviewer listed are all arXiv preprints and none of them are officially published. Nevertheless, below we list detailed comparison between our approach and each of the four references.  It shows that our results outperform all of them (we only list those experiments that overlap between our paper and the reference).
>
> The first reference (‘Learned Step Size Quantization’) is completely orthogonal to our work, and it is difficult to find any common part between our work and their paper.
>
> The two paper start from different views to analyze the quantization procedure. In the reference, the author proposes a quantizer with trainable step size, and improves training convergence by balancing the magnitude of step size updates with weight updates. The basic reasoning of the reference is that the ratio between update magnitude and parameter magnitude (the original value) should be similar for the learned step size and the learned weight. Meanwhile, to correct the impact of quantization precision, a gradient scale is introduced for the step size loss. In our paper, on the other hand, we start from the beginning by analyzing the training dynamics with mean field theory, which is a method from condensed matter physics but is widely adopted in literature on analysis of weight initialization, to propose some efficient training rules for a generic neural network. Based on our analysis, we investigate the existing quantization algorithms and found some procedure in typical quantization algorithms such as DoReFa violates the efficient training rules which leads to degenerated performance.
>
> The methods proposed in the two papers are also completely irrelevant. In the LSQ paper, the author focuses on training the step size, and scale the gradients. However, our method focuses on the scale of the weights themselves, through which we manage to control the training dynamics, and influence the gradients indirectly.
>
> The results of the LSQ also differs from ours drastically. Actually, the only common experiments for the two papers is ResNet-50 on ImageNet, while ours significantly outperform LSQ by as large as 0.8% for 3 bit case, as indicated in the following table.
>
>  Actually, all the four references show worse performance than our approach. We are open to include these references in the modified version, but it does not change our claim that we still achieve the state-of-the-art performance on model quantization, especially on challenging mobile-scenario networks.

---

> > ### Author Response · Authors · 2019-11-07
> > **Performance Comparison with the References Mentioned by the Reviewer**
> >
> > Reference 1: https://arxiv.org/pdf/1902.08153.pdf
> > -------------------------------------------------------------------------------------------------
> > |      Model      |   Method   |      Top-1 Valid Acc.     |   Top-5 Valid Acc.   |
> > -------------------------------------------------------------------------------------------------
> > |                        |                    |    2 bit    3 bit    4 bit    |  2 bit   3 bit   4 bit  |
> > -------------------------------------------------------------------------------------------------
> > |                        |       LSQ      |    73.7    75.8    76.7    |  91.5   92.7   93.2    |
> > |   ResNet-50   |                    |                                        |                                |
> > |                         | SAT (ours) |   74.1    76.6    76.9     |  91.7   93.1   93.3   |
> > -------------------------------------------------------------------------------------------------
> >
> > Reference 2: https://arxiv.org/pdf/1903.08066.pdf
> > ------------------------------------------------------------------------------------------------------
> > |      Model             |   Method   |      Top-1 Valid Acc.     |   Top-5 Valid Acc.   |
> > ------------------------------------------------------------------------------------------------------
> > |                               |       TQT      |           75.4 (8 bit)         |       92.3 (8 bit)       |
> > |      ResNet-50      |                    |                                        |                                 |
> > |                               | SAT (ours) |          76.9  (4 bit)         |      93.3 (4 bit)        |
> > ------------------------------------------------------------------------------------------------------
> > |                               |       TQT      |           71.1 (8 bit)         |       90.0 (8 bit)       |
> > |   MobileNet-V1   |                    |                                        |                                 |
> > |                               | SAT (ours) |          72.6  (8 bit)         |      90.7 (8 bit)        |
> > ------------------------------------------------------------------------------------------------------
> > |                               |       TQT      |           71.8 (8 bit)         |       90.6 (8 bit)       |
> > |   MobileNet-V2   |                    |                                        |                                 |
> > |                               | SAT (ours) |          72.5  (8 bit)         |      90.7 (8 bit)        |
> > ------------------------------------------------------------------------------------------------------
> >
> > Reference 3: https://arxiv.org/pdf/1905.11452.pdf
> > ------------------------------------------------------------------------------------------------------
> > |      Model             |   Method   |      Top-1 Valid Acc.     |   Top-5 Valid Acc.   |
> > ------------------------------------------------------------------------------------------------------
> > |                               |       DQ       |           70.59 (4 bit)        |              NA              |
> > |   MobileNet-V2   |                    |                                        |                                  |
> > |                               | SAT (ours) |          71.1  (4 bit)         |      90.0 (4 bit)         |
> > -------------------------------------------------------------------------------------------------------
> >
> > Reference 4: https://arxiv.org/pdf/1905.13082.pdf
> > ------------------------------------------------------------------------------------------------------
> > |      Model             |   Method   |      Top-1 Valid Acc.     |   Top-5 Valid Acc.   |
> > ------------------------------------------------------------------------------------------------------
> > |                               |       ref       |           66.46 (4 bit)        |              NA              |
> > |   MobileNet-V1   |                    |                                        |                                 |
> > |                               | SAT (ours) |          71.3  (4 bit)         |      89.9 (4 bit)        |
> > ------------------------------------------------------------------------------------------------------

---

### Public Comment · ~Steven_K._Esser1 · 2019-10-07
**Two points to clarify interpretation of results and novelty of technique**

1) Weight normalization methods have been previously shown to improve the performance of full precision networks (https://arxiv.org/abs/1602.07868), therefore it was surprising to see that the scores of full precision networks trained with the author's weight normalization procedure (SAT) were lower than scores reported on the same networks in other publications.  Do the authors have any ideas why this might have been the case?  Is it possible that the training hyperparameters used for the full precision case were not optimal?  For example, for a full precision network, the initial learning rate and weight decay values used seem a bit low.

Clarifying this is relevant because an alternative to the author's claim that SAT helps quantized networks outperform their full precision counterparts is that instead this weight normalization method improves the performance of both full precision and quantized networks, while not actually helping close the performance gap between them.  This alternative would also be an interesting result, particularly if it can be shown to out perform existing weight normalization methods.

2) The method by which the activation quantization parameter loss gradient is computed appears to be an adaptation of that used for LSQ in https://arxiv.org/abs/1902.08153.  It would be informative to clarify if any meaningful differences exist compared to this prior work.

---

> ### Author Response · Authors · 2019-11-07
> **Reply to the Reader**
>
> We appreciate the comment from the reader. However, it seems that the reader has some misunderstanding of our results.
>
> For the first point the reader mentioned, our results do show that weight normalization helps improve the performance of full precision networks, as demonstrated in Figure 2(b) of our paper. Weight normalization improves the performance of full precision networks from the red lines to the blue lines. The reason for worse results than those from the literatures can be that others use full precision results with original weights, while we use clamped weights, which is a more reasonable initial point for quantization (because they resemble the quantized weights more than those without clamping).
>
> Gradient calibration is not the most important contribution of our work, but compared with previous results (including the reference the reader provided), we give the detailed derivation of the result (in Appendix D), as well as experimental results on its impact (Figure 3).

---

### Author Response · Authors · 2019-11-07
**Novelty and contribution of our work**

We thank all reviewers for their reviewing effort and constructive comments to our work. All reviewers seem to have some doubts on the novelty of our work, so we reiterate our main contribution here for reference. In this paper, we investigate a problem that has been largely overlooked by most literature on quantization-aware training, that is, whether the accuracy drop of quantized neural networks comes from the reduced capacity, or inefficient training during the quantization procedure. To this end, we first investigate the condition of efficient training of a generic deep neural network, and discover that proper convergence of the network should follow two specific rules of efficient training. It is then demonstrated that deviation from these rules leads to improper training and accuracy reduction, regardless of the quantization level of weights or activations. To deal with this problem, a simple yet effective technique named scale-adjusted training (SAT) is proposed. We apply this method on a popular and typical quantization method PACT/Dorefa to showcase the effectiveness of the approach. The resulted quantized models significantly outperform the original PACT approach and validates our analysis. Violations of training rules also exists in other quantization approaches such as BinaryNet, XNORNet, Ternary Quantization, HWGQ as well as a differentiable quantization approach Darts Quant. We believe our approach can be used to boost the performance of such quantization algorithms as well.

References
BinaryNet: M. Courbariaux, et al., Binarized Neural Networks: Training Deep Neural Networks with Weights and Activations Constrained to +1 or -1. arXiv 2016.
XNORNet: M. Rastegari, et al., XNOR-Net: ImageNet Classification Using Binary Convolutional Neural Networks. ECCV 2016.
Ternary Quantization: C. Zhu, et al., Trained Ternary Quantization. ICLR 2017.
HWGQ: Zhaowei Cai, et al., Deep Learning with Low Precision by Half-Wave Gaussian Quantization. CVPR 2017.
Darts Quant: S. Uhlich, et al., Differentiable Quantization of Deep Neural Networks. arXiv 2019.

---

### Author Response · Authors · 2019-11-13
**Revised Version Submitted**

We appreciate the invaluable comments from the reviewers and readers, and we have submitted a revised version of our paper based on these comments. We hope the revised version makes our work clearer.

---

### Decision · Program_Chairs · 2019-12-19

**Decision:**

Reject

**Comment:**

The submission proposes methodology for quantizing neural networks.  The reviewers were unanimous in their opinion that the paper is not suitable for publication at ICLR.  Concerns included novelty over previous works, comparatively weak baseline comparisons, and overly restrictive assumptions.